# Neural Networks for Conversion of Simulated NMR Spectra from Low-Field to High-Field for Quantitative Metabolomics

**DOI:** 10.3390/metabo14120666

**Published:** 2024-12-01

**Authors:** Hayden Johnson, Aaryani Tipirneni-Sajja

**Affiliations:** 1Department of Biomedical Engineering, The University of Memphis, Memphis, TN 38152, USA; htjhnson@memphis.edu; 2Department of Biomedical Engineering, University of Houston, Houston, TX 77204, USA

**Keywords:** NMR spectroscopy, neural networks, metabolomics, transformer, low-field NMR

## Abstract

**Background:** The introduction of benchtop NMR instruments has made NMR spectroscopy a more accessible, affordable option for research and industry, but the lower spectral resolution and SNR of a signal acquired on low magnetic field spectrometers may complicate the quantitative analysis of spectra. **Methods:** In this work, we compare the performance of multiple neural network architectures in the task of converting simulated 100 MHz NMR spectra to 400 MHz with the goal of improving the quality of the low-field spectra for analyte quantification. Multi-layered perceptron networks are also used to directly quantify metabolites in simulated 100 and 400 MHz spectra for comparison. **Results:** The transformer network was the only architecture in this study capable of reliably converting the low-field NMR spectra to high-field spectra in mixtures of 21 and 87 metabolites. Multi-layered perceptron-based metabolite quantification was slightly more accurate when directly processing the low-field spectra compared to high-field converted spectra, which, at least for the current study, precludes the need for low-to-high-field spectral conversion; however, this comparison of low and high-field quantification necessitates further research, comparison, and experimental validation. **Conclusions:** The transformer method of NMR data processing was effective in converting low-field simulated spectra to high-field for metabolomic applications and could be further explored to automate processing in other areas of NMR spectroscopy.

## 1. Introduction

NMR spectroscopy is an indispensable tool across physical and life sciences with an incredibly broad range of applications across academia, healthcare, government, and industry; however, the expensive upfront cost, cost of maintenance, personnel requirements, and housing requirements of high-field magnets are barriers for many researchers that could potentially benefit from the NMR technology. Recent advances in magnet technology and NMR hardware have led to the development of a new generation of low-field (LF) NMR spectrometers that fit on the benchtop. These instruments are affordable with almost no maintenance cost, do not require dedicated lab personnel or cryogenic liquids, and the magnetic field extending beyond the device housing is negligible [1,2]. High-field (HF) instruments are very large and can take up a significant portion of a room, while benchtop instruments fit on a table and are portable and easy to operate—opening up possibilities for widespread use and point-of-care applications. Compared to HF NMR, LF NMR spectroscopy is less sensitive and has lower spectral resolution, making it harder to interpret spectra for metabolomics; thus, alternate methods may be required for optimal metabolite profiling. This research trains neural networks (NNs) to improve LF NMR signal quality for metabolite profiling by converting simulated LF spectra into their HF counterpart and will compare this to direct LF NMR metabolite quantification.

Numerous commonly used sequence-to-sequence NN models may prove useful for the task of converting LF spectra to HF spectra such as autoencoders, temporal convolutional neural networks, recurrent methods (like long short-term memory (LSTM) or gated recurrent units (GRUs)), or transformers [3]. Autoencoders employ encoder–decoder structures like densely connected autoencoders (DAEs), convolutional autoencoders (CAEs), and U-Nets [4] and find use in applications like signal denoising [5,6] or image processing [7]. Temporal convolutional networks (TCNs) gained popularity due to their impressive performance on sequence modeling tasks like forecasting [8,9] or time-series classification [10], which had generally employed CNNs or recurrent methods. Transformers are attention-based networks that have achieved impressive, state-of-the-art performance across sequence modeling tasks, especially in natural language processing tasks as exemplified by large language models like BERT and ChatGPT [3,11,12,13]. Transformers employ a self-attention mechanism to capture short and long-range dependencies within an input sequence; i.e., each member of the input sequence learns parameters relating it to all other members of the sequence and thus each member takes its context within the sequence into account [14]. Transformers are also attractive due to their high parallelizability compared to convolutional or recurrent methods—making them faster to train on modern graphics processing units (GPUs) [3].

This work implements DAE, CAE, U-Net, TCN, and transformer models for the task of transforming the spectral resolution of simulated LF 100 MHz NMR spectra of metabolite mixtures to HF 400 MHz spectra. Under the assumption that the entire spectra may potentially not be required simultaneously for successful conversion of LF to HF spectra, we also experiment with an approach that divides spectra into smaller spectral regions, which are independently fed into a model before being recombined into output spectra. Additionally, we train and test multi-layered perceptron (MLP) networks for both direct LF and HF quantification and compare this to first converting simulated LF spectra to HF prior to inputting into the model trained for HF quantification. In summation, this study introduces and validates a novel NN-based NMR data processing approach for converting simulated spectra from 100 MHz spectral resolution to 400 MHz.

## 2. Methods

### 2.1. Data Generation

Simulated LF (100 MHz) and HF (400 MHz) 1D-^1^H NMR spectra for 21 metabolites were downloaded from the human metabolome database (Maleic acid, histidine, 1-methylhistidine, acetic acid, creatine, glycine, formic acid, hypoxanthine, L-alanine, lysine, lactic acid, inosine, pyruvic acid, succinic acid, xanthine, creatinine, leucine, L-valine, NAD, niacinamide, and alpha-D-glucose). The simulated spectra in the HMDB were computed by predicting chemical shifts using a combination of machine learning and HOSE-code methods, determining coupling constants using empirical rules, and then spin matrix calculations were used to generate the spectra [15]. Simulated spectra largely resemble their experimentally acquired counterparts, with ^1^H chemical shifts generally differing by less than 0.15 ppm root mean squared error (RMSE) per spectrum [15]. Data processing was performed using Python (version 3.11.5) with the numpy library (version 1.24.3) for matrix operations and the nmrglue library (version 0.10) for NMR pre-processing operations (reading files, peak shifts, and Fourier transformations). To generate spectra, all 21 simulated spectra were scaled independently using scalars ranging uniformly from 1 to 50 and were summed to produce simulated mixture spectra with concentrations ranging from 1 to 50 mM. A singlet signal at 0.0 ppm was added to mimic a 3-(Trimethylsilyl)propionic-2,2,3,3-d 4 acid (TSP-d4) resonance of 2.96 mM in each spectrum to mimic a quantitative reference signal. Data augmentation was employed on each spectrum to produce a more varied dataset and mimic experimental signal variations by adding uniformly distributed noise (peak-to-peak maximum amplitude of ~0.1% of the quantitative reference peak height), randomly shifting metabolite peaks left or right (0–3.4 ppb), shifting baseline (up to ~0.6% of the quantitative reference peak height), and the addition of up to three randomly scaled singlets (using the acetic acid peak for a generic singlet) and up to three randomly scaled triplets (using the succinic acid signal for a generic triplet downloaded from the HMDB) at random chemical shifts. These steps were repeated 10,000 times with all 21 metabolites present in all spectra and 10,000 times with each metabolite having a 50% chance of being left out for a total of 20,000 spectra. This dataset was split 16,000:4000 for training and validation, respectively. A testing dataset of 4 simulated spectra was generated with all 21 metabolites present in all spectra. Several further example spectra, each with mean intensity noise added, were generated for testing analyte quantification including all 21 metabolites at 5 mM, all metabolites at 25 mM, and all metabolites at 50 mM. For all spectra in this study, only 46,000 data points from −0.32 ppm to 10.2 ppm corresponding to the signal-containing spectral regions were used for network training, testing, and validation.

Every spectrum generated using the above-described method was further generated following two alternative protocols. One dataset is generated with four times greater noise intensity added to LF than was added to HF spectra to simulate a lower signal-to-noise ratio (SNR) in LF spectra. The other dataset was generated with 87 metabolites rather than 21 (additional metabolites include: 2-Hydroxybutyric acid, 3-hydroxybutyric acid, alpha-ketoisovaleric acid, adenine, adenosine monophosphate, acetoacetic acid, dimethylglycine, citric acid, choline, ethanol, D-glucose, fructose 6-phosphate, glutathione, glycerol, fumaric acid, glutamic acid, L-tyrosine, phenylalanine, proline, L-threonine, L-asparagine, D-mannose, isoleucine, inosinic acid, serine, L-aspartic acid, isocitric acid, L-acetylcarnitine, oxoglutaric acid, myo-inositol, ornithine, NADP, oxalacetic acid, taurine, sarcosine, uridine 5′-monophosphate, uridine, 3-methyl-2-oxovaleric acid, L-arginine, acetylglycine, adenosine triphosphate, L-cysteine, glutamine, D-alpha-aminobutyric acid, ketoleucine, methionine, isovaleric acid, 3-hydroxyisovaleric acid, trimethylamine N-oxide, L-tryptophan, succinyl-CoA, fructose 1,6-bisphosphate, FADH, acetyl-CoA, FAD, ADP, NADH, phosphocreatine, nicotinic acid mononucleotide, acetone, caffeine, methanol, propylene glycol, itaconic acid, selenocysteine, and oxidized glutathione) and with four times greater noise for LF spectra relative to HF.

Three training and validation datasets were generated for metabolite quantification by MLP. These datasets were generated using the same data generation workflow as the above-described datasets (i.e., 21 metabolites, 21 metabolites with adjusted SNR, and 87 metabolites with adjusted SNR), with the only difference being no triplets were added in data generation. Three testing spectra, each with mean intensity noise added, were generated for assessing analyte quantification including all 21 (or 87) metabolites at 5 mM, all metabolites at 25 mM, and all metabolites at 50 mM.

To explore how the model performs on metabolites that it has not yet seen, a testing dataset of 50 spectra was generated using the workflow described above (varying noise, varying peak shift, varying baseline shift, and with SNR 4 times lower for 100 MHz spectra compared to 400 MHz). This unseen metabolite dataset contained 12 metabolites using simulated spectra downloaded from the HMDB for deoxyuridine, p-hydroxyphenylacetic acid, 3-methoxytryramine, dihydrobiopterin, adenosine, cyclic AMP, deoxyinosine, dopamine, cysteinylglycine, dihydrothymine, cytidine triphosphate, and dimethylamine.

### 2.2. Network Architectures, Training, Validation, and Testing

Pytorch (version 2.2.1) was used for data loading, model development, and training. All models were trained using LF 100 MHz spectra as input, HF 400 MHz spectra as the target, mean squared error (MSE) as a loss function, Adam as an optimizer using the default learning rate, and ReLU as the activation function. Models were trained for 300 epochs or until 25 epochs passed without a new best validation loss value achieved. All models were evaluated using MSE between the 100 ground-truth testing high-field spectra and predicted high-field testing spectra, and four testing output spectra were assessed visually against ground-truth 100 and 400 MHz spectra. The best model was further validated on 50 spectra of 12 metabolites—all of which were not seen by the model in training.

The DAE was instantiated with an encoder of densely connected layers consisting of 46000, 2000, and 200 nodes respectively (with 46,000 data points being the size of each input spectra), and the decoder was the reciprocal (200-2000-46000 nodes). The CAE was developed with a four convolutional layer encoder consisting of 16, 32, 64, and 128 kernels, respectively, and the decoder was the reciprocal (128-64-32-16 kernels). CAE kernels were of a size of 3 units with a stride and padding of 1. The U-Net model employed the same architecture as the CAE but with a skip connection between the 32 kernel layers. To reduce computational requirements, an alternate U-Net approach was tested where each input spectrum was separated into 46 bins of 1000 data points each before being fed into the model and subsequently concatenating after inference. This binned U-Net approach is referred to as UNet-Chunks for the remainder of this manuscript. The TCN implementation using the Pytorch framework is described by Bai et al. and found on GitHub (https://github.com/locuslab/TCN (accessed on 7 June 2023)) [16]. The TCN used a kernel size of two, dropout rate of 20%, and three temporal blocks with 25, 50, and 100 channels, respectively. ReLU was used for all activation functions.

For the transformer, an encoder-only transformer was followed by a simple linear layer—a minor modification of the transformer architecture described by Vaswani et al. [3]. The 46,000 data point spectra inputs were separated into 46 bins of 1000 data points each prior to model input, and model inputs were considered sequences of length 46 with 1000 features each. Inputs were passed to a linear layer converting features to 512 embeddings. No positional encodings were applied to these embeddings before passing them into a 6-layer transformer encoder with 8 attention heads, a feedforward dimension of 2048 nodes, and dropout applied at a rate of 10%.

MLPs with 46,000 input nodes, a 200-node hidden layer, and 21 (or 87 depending on the testing dataset) output nodes were trained for quantification of analytes at 100 MHz and 400 MHz using the same datasets used to train for LF-to-HF conversion. These models were assessed on the three example spectra with all metabolites at 5, 25, or 50 mM using mean absolute percent error (MAPE) as an accuracy metric, and these models were compared to the method of first converting LF spectra to HF using the transformer and then applying the HF-MLP for inference.

## 3. Results

The DAE, CAE, UNet, UNet-Chunks, TCN, and transformer model were all trained for the conversion of ^1^H-NMR spectra of 21 metabolite mixtures from 100 MHz to 400 MHz. All models besides the transformer performed poorly in the task of LF-to-HF conversion with testing MSEs ranging from 2 × 10^−4^ to 3 × 10^−4^ (MSE for all models listed in Table 1 below). The DAE with a testing MSE of 2 × 10^−4^ performed poorly on testing data by always predicting HF spectra with ~20 mM of each metabolite and with some peaks not matching up with ground-truth signals regardless of the input, as seen for four overlays of ground-truth (red) and predicted (blue) HF spectra in Figure 1, and in several zooms of one spectrum showed in Appendix A. The CAE, UNet, UNet-Chunks, and TCN models each achieved a testing MSE of 3.0 × 10^−4^ and produced unsuccessful conversions resulting in noisy spectra that did not resemble ground-truth signals. Four spectral overlays for the CAE HF predicted spectra and ground-truth testing spectra are shown in Figure 2 (also one spectrum and two zoomed regions for the CAE model are displayed in Appendix A, and four spectra for the UNet, UNet-Chunks, and TCN are shown in Appendix A respectively). The UNet, UNet-Chunks, and TCN models all performed similarly to the CAE.

The transformer was the only successful neural network architecture in the current study for LF-to-HF conversion. Figure 3 displays four converted HF spectra in blue overlaid with the ground-truth HF spectra in red—with each estimated HF spectrum closely resembling its respective ground-truth spectrum. A closer look at the converted resonances and how they compare to each model LF input and corresponding HF ground-truth spectrum are shown in Figure 4. The transformer achieved a low MSE of 6.5 × 10^−5^ for this conversion on 100 testing spectra.

After the successful conversion of simulated 100 MHz to 400 MHz spectra for 21 metabolites, the same transformer architecture was trained for converting spectra with the same 21 metabolites but with four times lower SNR in LF spectra. Results were highly similar to model performance before adding noise (testing MSE of 1.0 × 10^−4^), and four spectra overlays of estimated HF and ground-truth HF spectra are displayed as Appendix A.

The same transformer architecture was next trained for LF-to-HF conversion in spectra containing up to 87 metabolites and with four times lower SNR for LF spectra, and four output spectra overlay with ground-truth signals are displayed in Appendix A along with respective MSE values. A closer look at an individual spectrum prior to and after conversion, along with the ground-truth spectrum, is shown in Figure 5 below. The model performs well for many metabolites; however, some smaller SNR resonances are not as successfully converted as before in spectra with 21 metabolites (testing MSE of 1.0 × 10^−4^).

The transformers trained on 21 and 87 metabolites, respectively, with adjusted SNR were further tested on 50 spectra of 12 metabolites—all of which had not been seen in training. The model trained on 21 metabolites achieved an MSE of 2.2 × 10^−4^, and the model trained on 87 metabolites achieved an MSE of 1.2 × 10^−4^ . The lower MSE is corroborated in the visual comparison of predicted and ground-truth high-field spectra shown in Figure 6 and Figure 7, where it is shown that both models perform fairly poorly on the lower SNR multiplets in the zoom from ~2.8 to 3.4 ppm but the transformer trained on 87 metabolites performs better in the zoom from ~0.5 to 1.05 ppm. Appendix A shows the quantitative comparison of four spectra for the model trained with 21 metabolites, and S9 shows the quantitative comparison of four spectra for the model trained with 87 metabolites.

Representative testing spectra with all metabolites at 5, 25, or 50 mM were prepared in the same manner as the three datasets utilized in this study for comparing LF and HF metabolite profiling by neural networks in simulated spectra. MAPE values reflecting quantitative accuracy for all examples are shown in Figure 8 below (Appendix A lists these values in the Appendix A). For all three datasets (21 metabolites, 21 metabolites with adjusted SNR, and 87 metabolites with adjusted SNR), MLPs trained on LF spectra consistently achieved slightly higher accuracy for all three concentration testing spectra (ranging from ~22 to 104% lower MAPE for LF quantification) compared to the HF-MLP MAPE on HF validation spectra. The HF-MLPs were further tested on HF spectra produced by converting spectra from 100 MHz to 400 MHz using the appropriate transformer model (i.e., 21 or 87 metabolite model). In spectra with 21 metabolites, LF-to-HF conversion prior to quantification using the HF-MLP resulted in a ~80 to 153% drop in accuracy compared to direct LF quantification. The decrease in accuracy was greater for spectra with 87 metabolites at ~299 to 673%.

## 4. Discussion

The current study investigated the feasibility of converting low-field NMR spectra to higher magnetic field strength spectra using a neural network approach. This work compared DAE, CAE, UNet, TCN, and transformer neural networks for the sequence transduction task of converting 100 MHz ^1^H-NMR spectra to 400 MHz spectra. Additionally, MLPs were trained for analyte quantification directly at LF and HF for comparison with the model performing LF-to-HF conversion prior to quantification. All models were assessed on simulated spectra with up to 21 metabolites, and the most effective model, the transformer, was further assessed on two datasets with adjusted SNR for LF spectra and up to 21 or 87 metabolites, respectively. Only the transformer produced satisfactory sequence conversion, while the DAE, CAE, UNet, UNet-Chunks, and TCN models all overfit. This result was evident visually and was confirmed by lower MSE values achieved by the transformer model. The transformer reliably converted spectra with up to 21 metabolites for the original and adjusted SNR datasets. Increasing to highly complex 87 metabolites decreased the performance of the LF-to-HF conversion.

Interestingly, the direct quantification of HF spectra did not reveal higher accuracy compared to the direct quantification of LF spectra as we expected—with the LF quantification producing slightly more accurate estimations in our analysis for all three example spectra from each testing dataset. Despite not achieving as high accuracy metrics as the direct LF-MLP, the HF-MLP quantification of converted LF-to-HF spectra was reasonably accurate and thus further proved the quality of the LF-to-HF conversion, especially for spectra with 21 metabolites. The surprising result of higher accuracy for quantification directly in LF spectra could be because the lower resolution 100 MHz resonances are wider and thus each resonance interacts with a greater number of tunable model parameters. Our results are similar to research by Burger et al. who found little difference between the performance of 43, 60, 500, and 600 MHz spectrometers in the task of molecular weight determination [17]. It is possible that quantification methods other than an MLP network, such as peak deconvolution followed by concentration computation or using alternative neural networks, might affect the analyte quantification results seen in this study and lead to lower quantitative performance at lower field strengths. While our current study proves the feasibility of our proposed low-field to high-field conversion method, not performing analysis in experimentally acquired spectra is a limitation, and future work should include performing a similar analysis to this study with experimentally collected spectra and biological spectra to validate our findings in simulated spectra.

The results of the transformer models on spectra containing only metabolites not seen by the models during training revealed that the model trained on 87 metabolites outperformed the model trained on 21 metabolites; however, the model trained on 87 metabolites still struggled to convert some resonances. This result suggests that the more resonances of varied chemical structures the model sees in training, the better it may generalize to structures it did not see in training. Thus, the model and data generation workflow deployed in the current study might be best suited for a targeted analysis in well-defined samples (i.e., the samples being studied have a suite of expected metabolites and a model is trained specifically for these metabolites), and practitioners seeking to convert any given metabolite in less well-defined samples should generate training spectra covering a much larger number of metabolites and thus a much wider chemical space.

As simulated spectra were used, it is also possible that our comparison does not fully reflect what would be encountered in experimentally collected LF and HF spectra in terms of SNR. Further validation of this LF-to-HF procedure should take care to ensure realistic SNR for all spectra. Our study applied a linear SNR dependency as mentioned in several publications [18,19] (four times higher for 400 MHz compared to 100 MHz); however, it may be more accurate to scale SNR by the power of 3/2 [20,21] (eight times higher for 400 MHz). Additionally, other factors beyond field strength affect SNR (e.g., the probe) [22], which could further confound comparisons between instruments. Despite the SNR limitations, the transformers effectively converted the spectral resolution of LF spectra to 400 MHz. Future work should include conversion between spectra with greater differences in magnetic field strengths than the 100 to 400 MHz spectra used in this study.

The transformer was effective in LF-to-HF conversion, but it is likely that modifications could improve model performance. The training/validation dataset of 20,000 spectra is relatively low for a transformer, and it is very likely that increasing to hundreds of thousands or even more spectra would increase LF-to-HF conversion performance for the transformer, and potentially for the other models that did not perform well. As only one set architecture and set of parameters were tested per model in this study, it is possible that through hyperparameter optimization, performance could be improved for all models assessed in this study, which warrants future investigation. Performance may improve by testing different architectures like an encoder–decoder architecture, convolutional autoencoders, or exploring modified attention mechanisms like local attention or customized attention (e.g., input data points which correspond to a particular analyte attend data points from the same analyte, as well as data points corresponding to common interferences, plus some noisy and randomly selected data points). It is also possible that dataset modifications (e.g., peak alignment, varying training concentration distribution, denoising, etc.) could further improve LF-to-HF performance, which is beyond the scope of the current study. Results from the current study show that the conversion of low-field to high-field spectra is feasible, and future studies should investigate the potential performance-enhancing modifications outlined above.

The transformer in this study did not utilize positional encoding yet still achieved quality results, but preliminary experiments with positional encodings did not harm or improve performance. No positional encodings were necessary due to the highly ordered structure of the NMR data (i.e., 46 bins of 1000 features are transformed simultaneously and concatenated; thus, each member of the 46-member sequence remains in the same chemical shift order), or possibly because the model learned to convert individual bins. It is also likely that dataset modifications or hyperparameter optimization could alter model performance for improved LF-to-HF conversion.

The transformer use case in this study may not yet be proven necessary given the higher accuracy achieved for metabolites quantified directly in LF spectra; however, the overall signal transduction method could be effective in automating and increasing speed for a myriad of NMR data processing. Operations such as reference deconvolution, denoising/signal enhancement, and general processing (phase and baseline correction) could be implemented via a transformer. The transformer could also be used for metabolomics use cases like peak alignment or metabolite quantification. It could be used for other exploratory methods as well, such as increasing realism in simulated NMR spectra.

In conclusion, this study explored several NN architectures for the conversion of LF NMR spectra to HF spectra with the goal of improving LF NMR metabolomics, and the results found the transformer network to be the optimal model. The methods in this manuscript have produced promising results in the area of LF-to-HF conversion, and they represent an extensible approach to data processing by transformer, which could be applied in many other areas of NMR spectroscopy.

## Figures and Tables

**Figure 1 metabolites-14-00666-f001:**
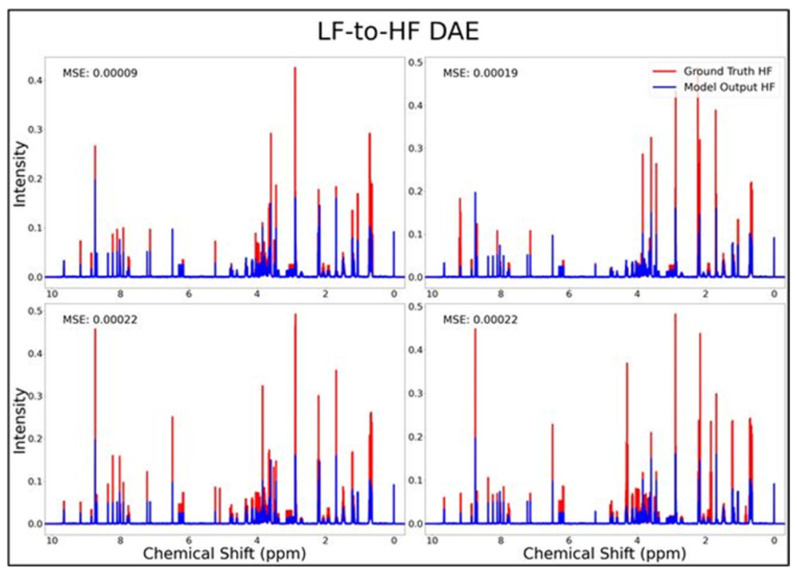
DAE performance results for 100 MHz to 400 MHz conversion of four testing spectra from the dataset of 21 metabolites. The ground-truth HF spectrum is shown in red overlaid with the corresponding LF-to-HF converted spectra in blue, and the MSE between spectra is displayed for each conversion. Abbreviations: LF = low-field; HF = high-field; DAE = densely connected autoencoder; MSE = mean-squared error.

**Figure 2 metabolites-14-00666-f002:**
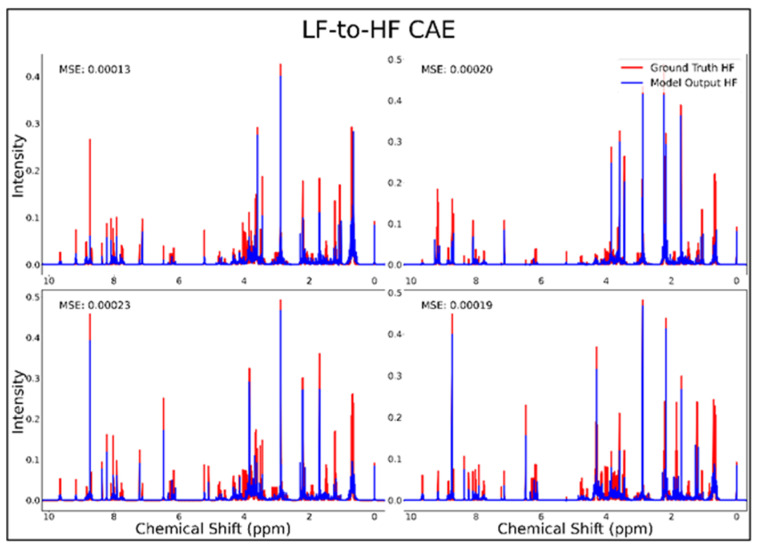
CAE performance results for 100 MHz to 400 MHz conversion of four testing spectra from the dataset of 21 metabolites. The ground-truth HF spectrum is shown in red overlaid with the corresponding LF-to-HF converted spectrum in blue, and the MSE between spectra is displayed for each conversion. Abbreviations: LF = low-field; HF = high-field; CAE = convolutional autoencoder; MSE = mean-squared error.

**Figure 3 metabolites-14-00666-f003:**
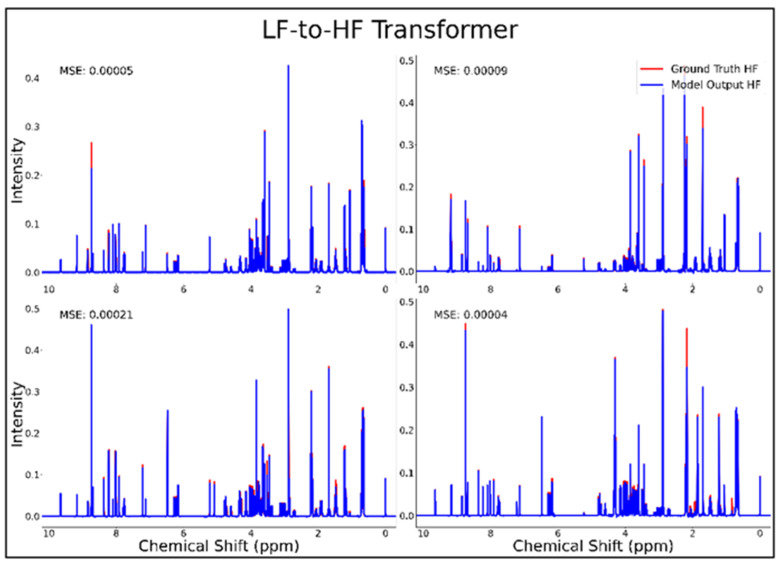
Transformer performance results for 100 MHz to 400 MHz conversion of four testing spectra from the dataset of 21 metabolites. The ground-truth HF spectrum is shown in red overlaid with the corresponding LF-to-HF converted spectrum in blue, and the MSE between spectra is displayed for each conversion. Abbreviations: LF = low-field; HF = high-field; MSE = mean-squared error.

**Figure 4 metabolites-14-00666-f004:**
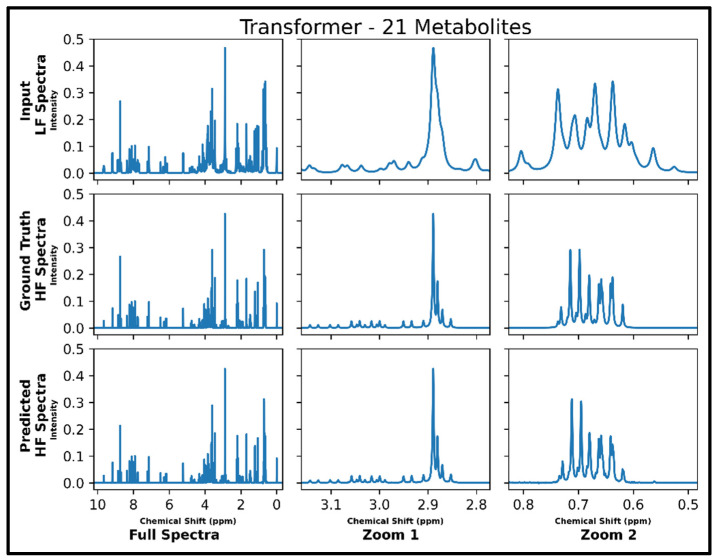
Transformer performance results for 100 MHz to 400 MHz conversion of one testing spectrum from the dataset of 21 metabolites. The top row shows the input LF spectrum, the middle row shows the ground-truth HF spectrum, and the bottom row shows the predicted HF spectrum. The left column shows the full spectra, and the middle and right columns show zoomed-in portions of the same spectra. Abbreviations: LF = low-field; HF = high-field.

**Figure 5 metabolites-14-00666-f005:**
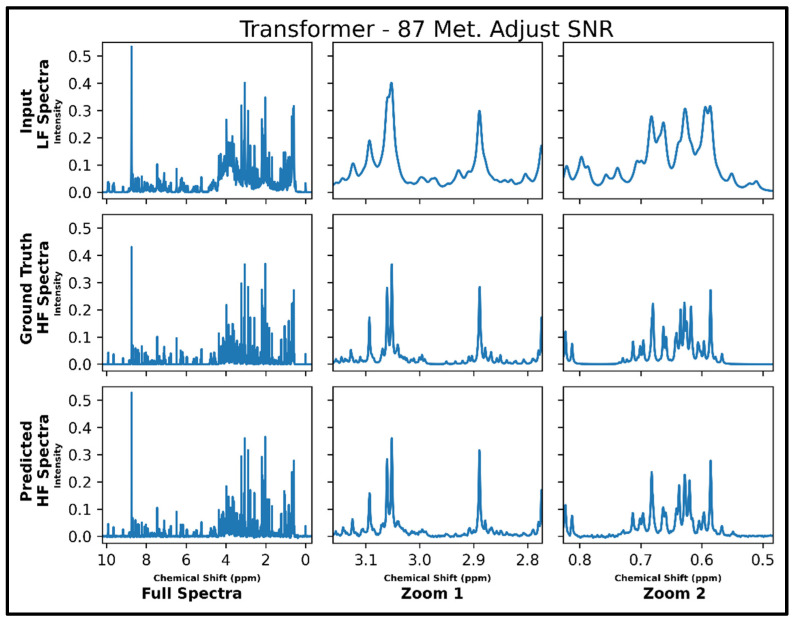
Transformer performance results for 100 MHz to 400 MHz conversion of one testing spectrum from the dataset of 87 metabolites with adjusted SNR (four times higher noise in 100 MHz compared to 400 MHz spectra). The top row shows the input LF spectrum, the middle row shows the ground-truth HF spectrum, and the bottom row shows the predicted HF spectrum. The left column shows the full spectra, and the middle and right columns show zoomed-in portions of the same spectra. Abbreviations: LF = low-field; HF = high-field.

**Figure 6 metabolites-14-00666-f006:**
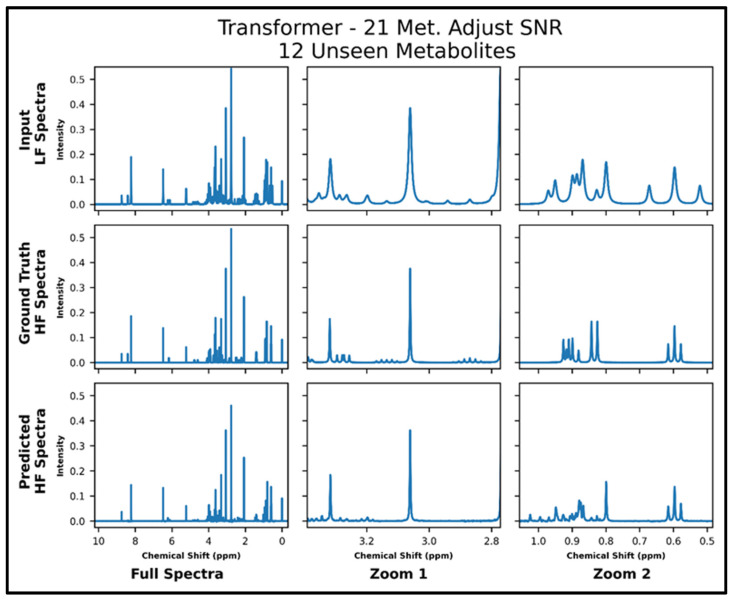
Transformer (trained with 21 metabolites and four times lower SNR for low-field spectra) performance results for 100 MHz to 400 MHz conversion of one test spectrum from the dataset of 12 metabolites not seen in training. The top row shows the input LF spectrum, the middle row shows the ground-truth HF spectrum, and the bottom row shows the predicted HF spectrum. The left column shows the full spectra, and the middle and right columns show zoomed-in portions of the same spectra. Abbreviations: LF = low-field; HF = high-field; Met. = Metabolites.

**Figure 7 metabolites-14-00666-f007:**
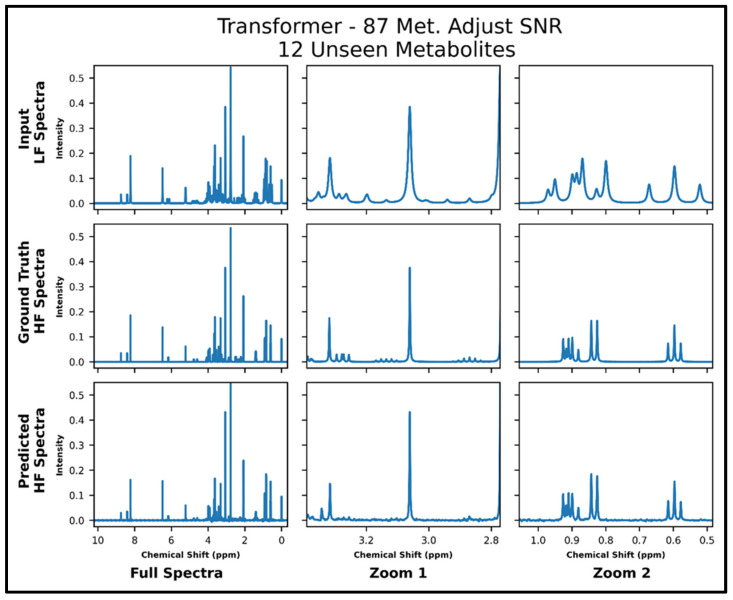
Transformer (trained with 87 metabolites and four times lower SNR for low-field spectra) performance results for 100 MHz to 400 MHz conversion of one test spectrum from the dataset of 12 metabolites not seen in training. The top row shows the input LF spectrum, the middle row shows the ground-truth HF spectrum, and the bottom row shows the predicted HF spectrum. The left column shows the full spectra, and the middle and right columns show zoomed-in portions of the same spectra. Abbreviations: LF = low-field; HF = high-field; CAE = convolutional autoencoder; Met. = Metabolites.

**Figure 8 metabolites-14-00666-f008:**
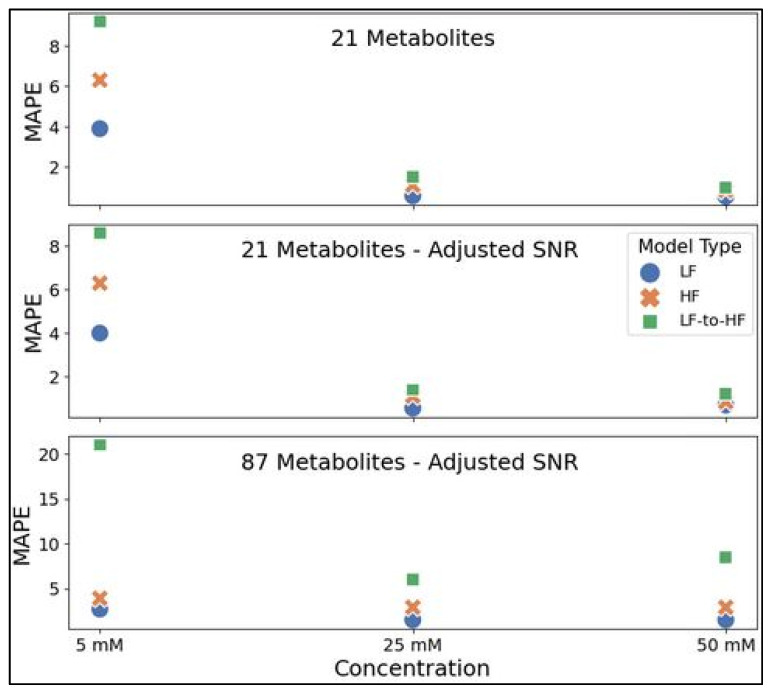
Performance for direct LF spectra quantification using the LF-MLP (blue circle), direct HF quantification using the HF-MLP (orange ‘x’), and LF-to-HF conversion prior to quantification using the HF-MLP (green square). MLPs were trained using either the 21 metabolites (**top panel**), 21 metabolites with adjusted SNR (**middle panel**), or 87 metabolites with adjusted SNR (**bottom panel**) datasets and each model was assessed for quantification accuracy using MAPE as a metric on 3 testing spectra with all trained metabolites (i.e., 21 or 87 metabolites) at 5 mM, 25 mM, and 50 mM. Abbreviations: HF = high-field; LF = low-field; SNR = signal-to-noise ratio; MAPE = mean absolute percent error.

**Table 1 metabolites-14-00666-t001:** Neural network performance metrics for 100 MHz to 400 MHz conversion of 10 NMR spectra. MSE between ground-truth HF spectra and predicted HF spectra are displayed for eight models.

**Model**	DAE	CAE	UNet	UNet-Chunks	TCN	Transformer- 21 Metabolites	Transformer- 21 Met. Adjust SNR	Transformer- 87 Met. Adjust SNR
**MSE**	2.0 × 10^−4^	3.0 × 10^−4^	3.0 × 10^−4^	3.0 × 10^−4^	3.0 × 10^−4^	6.5 × 10^−5^	1.0 × 10^−4^	1.0 × 10^−4^

Abbreviations: MSE = mean-squared error; HF = high-field; LF = low-field; DAE = densely connected autoencoder; CAE = convolutional autoencoder; TCN = temporal convolutional network; Met. = metabolites; SNR = signal-to-noise ratio.

## Data Availability

The code to reproduce the results presented in this study are available on Github (https://github.com/tpirneni/LF-to-HF-NMR).

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
