# Peer review of "Neural Networks for Conversion of Simulated NMR Spectra from Low-Field to High-Field for Quantitative Metabolomics"

_metabolites, 2024, doi:10.3390/metabo14120666_

Round 1
Reviewer 1 Report
Comments and Suggestions for Authors
The paper looks at a comparison of different neural network architectures for converting simulated 100-MHz NMR spectra to 400-MHz spectra for comparison. The paper uses what appears to be entirely simulated dataset from a very small subset of metabolites. Its compares methods under this scenario to some extent, but there is little to no network optimisation which significantly undermines any conclusions around the best method unfortunately. While the conclusions based on this limited study seems valid, I am not convinced of the purpose and therefore contribution of such a study. The simulated scenario seems unrealistic compared to what the real scenario is. There are far less metabolites in the study than I assume there would be in practise and the study does not consider the scenario of metabolites that are not in the training data. There is no consideration as it how realistic their generated NMR spectra are or how realistic the amount of training data might be if this was to be applied to real data. There is no validation at all on real data.
Unfortunately, while I found the paper interesting and well written, these concerns lead me to believe this paper is not suitable for publication at this stage.
General Comments
I think you need to make it more clear this is a simulated data study. Potentially adding ‘simulated’ into the title and adding it more often in the abstract and introduction
Paragraph overlapping pages 1 and 2 – Appears to be lacking references to the core papers which invented most of these concepts, e.g. transformers, U-Nets, BERT (You do eventually cite the transformers paper, but a few sentences late).
Paragraph overlapping pages 1 and 2 – it might be worth adding a brief idea of the concept of attention and self attention
Last paragraph of introduction – you need to be clearer that this is all done on simulated data, otherwise it is misleading.
Methods (data generation) – Does HMDB have simulated data? If so, a bit more information is needed about what it is, how it was generated, and how it was validated. I say this as the quality of the simulated data seems vital to validating the utility of your approach.
Methods (data generation) – you have a lot of simulated NMR spectra from only a small amount of metabolite spectra (21). How realistic is this, surely there are lots more metabolite spectra available? How does the method do on unseen spectra?
Methods (Network Architectures, Training, and Validation) – the differences in DAE nodes seems quite wildly different. What is the justification here? Or is 46000 is the number of inputs? (are these normally called nodes?)
Results (paragraph 1) – please clarify in the text that Figure 1 is on the validation set. Currently it read as if I am going to see overfitting in Figure 1 which there isn’t
Results (paragraph 1) – You say methods gave ‘unsuccessful conversions’ but don’t really say how this was quantified. While I can see what you mean in the figures, I think you need some quantification here. Can you get validation / test MSEs (whichever one you did not use in training loss. I know people define them both ways round – but I think you actually have test and validation the wrong way around, so it might be worth correcting or defining https://en.wikipedia.org/wiki/Training,_validation,_and_test_data_sets ).
Results (bottom page 6) – I worry that the models struggle on 87 spectra which surely in reality there are far far more? Why are you convinced this might work in practise?
Discussion – Not doing any form of network tuning seems a huge issue here.
I am confused as to why this method might work in practise when it has only been trained on a small, simulated dataset? Surely to work in practise you would need to train and validate the methods on pairs of real 100mhz-400mhz spectra, or at least use a lot more metabolite spectra. The method does not taking into account what happens in you see spectra outside of the trained spectra. I guess I am not sure what the paper contributes if you don’t use real data and it has no chance of being used in practise. You don’t even know the transformer would be the best method in practise as the data comes from such an unrealistic scenario with unoptimized networks.
Author Response
"Please see the attachment."

Reviewer 2 Report
Comments and Suggestions for Authors
In the current study, the authors present an conversion of low-field to high-field metabolite spectra data using neural network methods, yielding promising results that could be highly valuable for metabolomics research.
In this paper, the authors specifically used a standard dataset, but it would be beneficial if they also applied biological samples to simulate the low-field to high-field spectra.
Can this method also be used to assign the peaks observed in the spectra?
Overall, this study could be highly useful for researchers and may become a cost-effective approach in the future.
Author Response
"Please see the attachment."

Reviewer 3 Report
Comments and Suggestions for Authors
In this study, the authors evaluated and compared deep learning models, including DAE, CAE, U-Net, U-Net-Chunks, TCN, and transformer architectures, to convert low-field NMR spectra into high-field spectra for metabolite quantification. This approach offers an interesting method to generate high-resolution NMR data from lower-resolution spectra. However, I have several questions that need to be addressed:
1. Is the deep learning-based approach still feasible if the frequency gap between LF and HF spectra increases? Will the model still be able to accurately map LF to HF data if too much information is lost?
2. Given that the training data is simulated using deterministic algorithms, how well do these simulations reflect real-world conditions, even with the application of data augmentation?
3. Were ReLU activation functions used for all layers in the network architectures?
4. A quantitative metric should be used to assess the predicted spectra compared to the ground truth, rather than relying solely on visual inspection.
5. In Fig. 6, if the quantification using LF data is better than with HF data, what is the benefit of converting LF data to HF data?
Comments on the Quality of English Language
The quality of English language is acceptable.
Author Response
"Please see the attachment."

Round 2
Reviewer 1 Report
Comments and Suggestions for Authors
Thank you for your detailed response to my queries, I hope they were helpful and not too misguided. While I still think its important to test this on real data, your arguments make sense for this kind of paper
Reviewer 3 Report
Comments and Suggestions for Authors
The authors have addressed my questions. I have no further questions.
Comments on the Quality of English Language
The quality of English language is acceptable.